# Effect of Intrasilicone Bevacizumab Injection in Diabetic Tractional Retinal Detachment Surgery: A Retrospective Case-Control Study

**DOI:** 10.3390/jcm9103114

**Published:** 2020-09-26

**Authors:** Seung Kook Baek, Min-Woo Lee, Young-Hoon Lee

**Affiliations:** Department of Ophthalmology, Konyang University College of Medicine, Daejeon 35365, Korea; backkka@hanmail.net (S.K.B.); bogus1105@gmail.com (M.-W.L.)

**Keywords:** bevacizumab, diabetic retinopathy, silicone oil, vitrectomy

## Abstract

Tractional retinal detachment (TRD) causes visual loss in diabetes mellitus patients. Silicone oil can be used as a tamponade to repair retinal detachment; however, intrasilicone injection is challenging. We aimed to evaluate the effect of intrasilicone bevacizumab injection in TRD surgery. This was a single-hospital, retrospective, case-control study of 44 patients (46 eyes). We reviewed medical histories and ophthalmic examination results. We administered silicone oil to 26 eyes (group I), and a combination of silicone oil and intravitreal bevacizumab injection to 20 eyes (group II). The main outcome measures were the logarithm of the minimum angle of resolution (logMAR) visual acuity and central macular thickness. Mean change in logMAR visual acuity was larger (*p* = 0.029) in group II (−0.99 ± 0.73) than in group I (−0.56 ± 0.80), 12 months postoperatively. Compared to group I, group II exhibited a lower mean (471.54 ± 120.14 μm vs. 363.40 ± 59.57 µm, respectively; *p* = 0.001), and mean change (−22.39 ± 203.99 μm vs. −72.40 ± 139.35 µm, respectively; *p* = 0.027), in central macular thickness, 1 month postoperatively. Intrasilicone bevacizumab injection immediately after vitrectomy may rapidly reduce central macular thickness and increase final visual acuity. Prospective studies are necessary to demonstrate long-term safety and efficacy.

## 1. Introduction

Tractional retinal detachment (TRD) in proliferative diabetic retinopathy (PDR) is a major cause of severe visual loss in patients with diabetes mellitus [1]. TRD with recent macular involvement indicates the need for vitrectomy, which remains a challenging procedure despite improvements in surgical instrumentation and techniques, and the development of medication targeting vascular endothelial growth factor (VEGF) [2].

Silicone oil (SO) is transparent, chemically inert, has a low specific gravity and high viscosity, and can be used as a substitute for the vitreous [3]. It has been extensively used as a tamponade for the repair of complicated retinal detachment caused by PDR, proliferative vitreoretinopathy (PVR), and other proliferative vitreoretinal diseases [4,5]. However, neovascularization of the retina or iris can recur even after aggressive vitrectomy combined with an SO tamponade [6]. VEGF plays an important role in the progression of several ocular diseases, and it is associated with neovascularization and increased vascular permeability [7]. However, in the presence of SO, the concentration of medication injected into the posterior segment of the eye is markedly different from that in a vitreous-filled eye, or from that in an eye with fluid infusion following vitrectomy [8]. According to Grzybowski et al. [9], injections into an eye filled with SO can be very challenging. There have been reports of satisfactory results following intravitreal triamcinolone acetonide injection [10] and the injection of VEGF inhibitors [11,12,13] into an SO-filled eye. In addition, Alishiri et al. [14] recently reported favorable results with the simultaneous usage of bevacizumab and SO following a vitrectomy in diabetic patients with TRD. However, in some cases, after SO injection in PDR surgery, macular edema remains in SO-filled eyes [15]. These results are thought to have an effect on visual acuity after surgery. It is a widely known fact that macular edema is relieved again when anti-VEGF or steroid injection is administered. However, due to safety issues in the SO-filled eye, a clear conclusion has not been established.

The purpose of the present study was to investigate the differences between SO-filled eyes with and without bevacizumab injection after vitrectomy for diabetic patients with TRD. In addition, we would like to consider the effects and advantages of intrasilicone bevacizumab injection immediately after vitrectomy.

## 2. Methods

### 2.1. Patients

This retrospective observational study was approved by the institutional review board (KYUH 2019-05-029) and all procedures adhered to the principles of the Declaration of Helsinki. We reviewed the charts of the PDR patients with TRD that involved the macula, who underwent a vitrectomy from January 2015 to November 2018. Among these patients, we enrolled the patients who completed the surgery with SO injection and SO removal. All patients were followed-up for more than one year. We reviewed the medical history of the enrolled patients and conducted an ophthalmic examination, including measurement of best-corrected visual acuity (BCVA), intraocular pressure (IOP), slit-lamp biomicroscopy, dilated fundus examination, and fundus photography. Optical coherence tomography (OCT) was conducted using a Heidelberg Spectralis^®^ imaging platform (Spectralis; Heidelberg Engineering, Inc., Heidelberg, Germany) prior to surgery. Patients were excluded if they had glaucoma, rhegmatogenous retinal detachment, retinal disease other than diabetic retinopathy (DR), high myopia with an axial length ≥ 27.0 mm, or a history of non-cataract intraocular surgery.

### 2.2. Surgical Technique and Postoperative Managements

All procedures were performed by the same surgeon (YH Lee), consisting primarily of 25-gauge vitrectomy using the Constellation^®^ Vision System (Alcon Laboratories Inc., Fort Worth, Texas, USA). Simultaneous cataract surgery was performed on cataracts with grade 2 or higher, according to the Lens Opacities Classification System (LOCS) III classification, during the first vitrectomy or SO removal. All patients received intravitreal bevacizumab (IVB) injections one day or one week before the surgery. After vitrectomy, the fibrovascular membrane was carefully removed by delamination and segmentation, and retinectomy was performed where the surgeon deemed necessary. Fluid–air exchange, endolaser photocoagulation, and SO (1000 centistokes) tamponade were performed in all cases. At the end of the surgery, 1.25 mg/0.05 mL bevacizumab (Avastin; Genentech, Inc., South San Francisco, CA, USA) was injected into the SO-filled vitreous cavity per the surgeon’s discretion. After surgery, all patients self-administered moxifloxacin (Vigamox^®^, Alcon Laboratories, Inc.) and 1% prednisolone acetate ophthalmic solution (Pred Forte^®^, Allergan, Inc., Irvine, CA, USA) four times daily for one week, followed by levofloxacin and 0.1% fluorometholone ophthalmic solution (Fluorometholone^®^, Taejoon Pharmaceutical Co., Ltd., Seoul, Korea) for three weeks.

### 2.3. Data Collection

Patients underwent postoperative ophthalmic examinations at 1 day, 1 week, and 1 month before SO removal, as well as at 1 week, 1 month, 2 months, and 12 months after SO removal. BCVA measurement and OCT were conducted during each of these examinations, in order to compare pre- and postoperative BCVA and central macular thickness (CMT) values. BCVA was measured using the Snellen chart, the values of which were converted to logMAR (logarithm of minimum angle of resolution) visual acuity (VA) scores for statistical analysis. CMT was defined as the average value of macular thickness in the area (with a 1-mm diameter) around the fovea centralis.

### 2.4. Statistical Analysis

Statistical analyses were performed using SPSS 21.0 (IBM Corp., Armonk, NY, USA; RRID:SCR_002865). Wilcoxon signed-rank tests were used to compare pre- and postoperative values within each group. Fisher’s exact tests and Mann–Whitney *U*-tests were used for between-group comparisons. In all cases, the threshold for statistical significance was defined as *p* < 0.05.

## 3. Results

### 3.1. Patient Demographics and Clinical Characteristics

We recruited 44 patients (46 eyes) in this study. Of these, 26 eyes received an SO tamponade without IVB injection (group I) and 20 eyes received both an SO tamponade and IVB injection (group II). There were no statistically significant between-group differences in terms of age, sex, laterality, hypertension, duration of diabetes mellitus, history of panretinal photocoagulation, and number of preoperative IVB injections. However, there was a statistically significant difference in preoperative glycated hemoglobin level (HbA1c) (group I, 8.17 ± 0.91% vs. group II, 9.56 ± 2.61%; *p* = 0.022). No statistically significant difference existed between groups regarding preoperative intraocular pressure (IOP), logMAR VA, or CMT. In terms of the vitrectomy, there was no statistically significant difference in the number of cataracts removed, the volume of SO injected, or the interval between SO injection and removal between groups (Table 1).

### 3.2. Visual Outcomes

Both groups exhibited improvements in mean logMAR VA 1 month after SO removal (group I, *p* = 0.031; group II, *p* = 0.003) and 12 months postoperatively (group I, *p* = 0.003; group II, *p* < 0.001) compared to the preoperative status. However, there were no statistically significant between-group differences in logMAR VA 1 month postoperatively, immediately before SO removal, 1 month after SO removal, or 12 months postoperatively.

Mean change in logMAR VA in group I was −0.13 ± 0.77, −0.13 ± 0.66, −0.42 ± 0.72, and −0.56 ± 0.80 at 1 month postoperatively, immediately before SO removal, 1 month after SO removal, and 12 months postoperatively, respectively. The mean change in logMAR VA in group II, for the same time points, was −0.38 ± 0.88, −0.24 ± 0.76, −0.68 ± 0.77, and −0.99 ± 0.73, respectively. There was a statistically significant between-group difference in terms of mean change in logMAR VA at 12 months postoperatively (*p* = 0.029) (Table 2, Figure 1).

### 3.3. Anatomic Outcomes

Mean CMT tends to decrease over time in both groups. In group I, there were no statistically significant differences in mean CMT from the preoperative status to 1 month postoperatively, or to immediately before SO removal. However, there were statistically significant decreases from the preoperative status to 1 month after SO removal (*p* = 0.020) and to 12 months postoperatively (*p* = 0.011). In group II, there were statistically significant decreases in mean CMT from the preoperative status to 1 month postoperatively (*p* = 0.047), to immediately before SO removal (*p* = 0.012), to 1 month after SO removal (*p* < 0.001), and to 12 months postoperatively (*p* < 0.001). Additionally, there were statistically significant between-group differences in mean CMT at 1 month postoperatively (*p* = 0.001), but not immediately before SO removal (*p* = 0.399), 1 month after SO removal (*p* = 0.092), or 12 months postoperatively (*p* = 0.424).

Mean change in CMT also shows that the change increases as time passes in both groups. There was a statistically significant between-group difference in mean change at 1 month postoperatively (*p* = 0.027), but not immediately before SO removal, 1 month after SO removal, or 12 months postoperatively (Table 2, Figure 2).

### 3.4. Postoperative Complications and Other Outcomes

There was no statistically significant difference in postoperative complications (IOP increase, vitreous hemorrhage, fibrovascular membrane proliferation) and postoperative HbA1c (12 months postoperatively) between the two groups. However, the number of IVB injections required after SO removal due to macular edema and increasing retinal microhemorrhages was statistically significantly higher in group I than in group II (23.1% vs. 0.0%, respectively; *p* = 0.025) (Table 3).

## 4. Discussion

TRD is an advanced-stage form of DR that occurs when contractile elements in the vitreous and neovascular growth tissue cause detachment of the neurosensory retina [2,16]. Development of DR, oxidative stress, and retinal ischemia lead to the upregulation of angiogenic factors, particularly VEGF, and several chemokines [17]. IVB injections prior to vitrectomy can reduce active neovascularization [18,19,20], intraoperative bleeding [21,22,23,24,25], postoperative bleeding [22,23], and duration of surgery [19,25,26] in diabetic patients with TRD. However, the concentration of angiogenic factors, such as VEGF, increase in response to surgical trauma and inflammation in DR patients [27,28]. Additionally, in patients with chronic disease such as PDR, inflammatory cytokines are produced in peripheral blood monocular cells (PBMCs), which are associated with VEGF level elevation [29]. High VEGF concentration following surgery may alter the integrity of retinal blood vessels, which may increase postoperative bleeding, persistent macular edema, and retina, iris, and disc neovascularization. In order to reduce this surge in VEGF levels, it has been reported [30,31,32] that IVB injections administered at the end of the PDR vitrectomy improved the surgical outcome.

The abovementioned results suggest that both preoperative and postoperative IVB injections can be used to reduce postoperative complications. After vitrectomy for TRD, SO is injected in some cases and maintained for a prolonged period. Intravitreal injections are not usually performed in an SO-filled eye, as the effective concentration of injected substances is unpredictable in such eyes [9]. However, triamcinolone acetonide [10,33], dexamethasone implant [34], and bevacizumab [12,13,14] have been injected in SO-filled eyes to improve surgical outcome.

Our retrospective study was conducted on diabetic patients injected with SO during surgery for TRD involving the macula, some of whom did not receive bevacizumab injection (group I), and some of whom did (group II). Chronic hyperglycemia causes pathologic changes in the retina and vitreous which leads to progressive deterioration, including TRD [35]. Following vitrectomy, uncontrolled blood glucose levels have been associated with undesirable visual outcomes, such as postoperative bleeding [36]. Thus, careful diabetic control is vital. In the present study, preoperative HbA1c was statistically significantly higher (*p* = 0.022) in group II (9.56 ± 2.61%) than in group I (8.17 ± 0.91%). Patients were referred to an endocrinologist for postoperative glycemic control. Thereafter, HbA1c levels of both groups decreased, with no statistically significant difference between groups.

IVB injection 1-14 days prior to surgery has been demonstrated to improve the ease of surgery in complex cases of TRD or vitreous hemorrhage [21]. However, others have reported that TRD progressed after preoperative IVB injection [37,38]. In our study, IVB was injected either 1 week or 1 day before TRD surgery, and there was no evidence of fibrovascular membrane proliferation in the surgical field.

In the presence of an SO tamponade, the concentration of medication injected into the posterior segment of the eye is markedly different compared to that in a vitreous-filled eye, or to that in a fluid-infused eye following vitrectomy [8]. SO acts as a barrier to the migration of angiogenic factors, although, due to the buoyancy of SO, the inferior retina remains in contact with aqueous [5,39,40]. Angiogenic factors, such as VEGF, can become trapped between silicone bubbles and the retina, which can lead to neovascularization or proliferation of preretinal or subretinal fibrous membranes [39,41]. In the present study, we observed postoperative complications. In group II, there were no developments of fibrovascular membranes or vitreous hemorrhages; however, there were two cases of increased IOP. In group I, there were two cases of fibrovascular membrane proliferation and one case of vitreous hemorrhage; however, there were no cases of increased IOP. There was no statistically significant difference between the two groups in terms of postoperative complications, and none of the patients required reoperation. We hypothesize that bevacizumab may have accumulated in the meniscus below the SO due to gravity. Any VEGF isolated in SO bubbles may therefore have reacted with the bevacizumab, reducing the likelihood of postoperative fibrovascular membrane proliferation or vitreous hemorrhage. However, further study is needed in order to clarify the exact mechanism involved.

Mean logMAR VA exhibited statistically significant improvement 1 month after SO removal and 12 months after vitrectomy in both groups. Although the mean change of logMAR VA was statistically significantly larger in group II 12 months after surgery, such a between-group comparison is not expected to be of clinical significance due to the variation in cataracts between patients.

In the present study, 61.5% of patients in group I and 80.0% of patients in group II underwent cataract removal during the vitrectomy, several patients underwent cataract surgery at the time of SO removal, and we did not exclude patients with preoperative pseudophakia from the study. This variation in cataracts between patients is one of the limitations to our study, as it may have led to selection bias. However, since there was no statistically significant difference between the initial visual acuity and the ratio of simultaneous cataract surgery between the two groups, it is thought that the effect of the selection bias on the study was insignificant. Through prospective studies, it may be necessary to discuss whether intrasilicone bevacizumab injection leads to better visual outcomes under tight control for cataracts.

There may be uncertainty about retinal toxicity when IVB is injected into a SO-filled eye. Salman [13] reported the regression of neovascularization in 12 eyes upon IVB injection (1.25 mg/0.05 mL). Salman also reported a subsequent increase in mean logMAR VA, and decreasing macular and corneal edema six months after injection. Falavarjani KG et al. [12] injected 2.5 mg/0.1 mL of bevacizumab into SO-filled eyes and observed no retinal toxicity or other complications. Our findings were in line with the abovementioned studies, as we determined that mean logMAR VA improved 12 months after IVB injection in SO-filled eyes. Therefore, we do not consider IVB injection (1.25 mg/0.05 mL) into a SO-filled eye to be toxic to the retina.

Xu et al. [42] performed vitrectomy and SO injection with IVB (1.25 mg/0.05 mL) in rabbits. Over 24–72 h, the bevacizumab gradually migrated into the vitreous fluid between the SO and the retinal surface, and after 48 h, the visible bevacizumab droplets had disappeared from the oil phase in over 90% of the eyes. Xu et al. [42] posited that, as a result, bevacizumab distribution was delayed, resulting in stable levels of bevacizumab in the ocular tissues. This could, in turn, lead to fewer systemic physiologic changes and reduced side effects.

In the present study, mean CMT exhibited a statistically significant decrease in both groups 1 month after removal of SO and 12 months after surgery. However, only in group II was there also a statistically significant decrease in mean CMT 1 month after the surgery and immediately before SO removal. Additionally, there were statistically significant between-group differences in mean CMT 1 month postoperatively, and in mean change of CMT 1 month postoperatively. We posit that the reason for this difference was the entrapment of bevacizumab after SO injection, which prolonged the exposure of the retina to bevacizumab. This would, in turn, prolong the inhibitory and antiangiogenic effect of bevacizumab.

In our study, there was an increased need for IVB injection due to macular edema and retinal microhemorrhage after SO removal in group I when compared with group II (23.1% vs. 0.0%, *p* = 0.025). This may be explained by the increased pre- and postoperative mean CMT in group I when compared with that of group II. Therefore, further study is required to determine whether the need for postoperative IVB injection is truly reduced due to intraoperative IVB injection.

In conclusion, we have demonstrated that intrasilicone bevacizumab injection during TRD surgery in diabetic patients is an effective technique for rapid CMT reduction and improved final visual acuity. Our study therefore supports the role of intrasilicone bevacizumab injection as a valuable option in diabetic TRD surgery. However, future randomized prospective studies are necessary to evaluate the efficacy and safety of this technique.

## Figures and Tables

**Figure 1 jcm-09-03114-f001:**
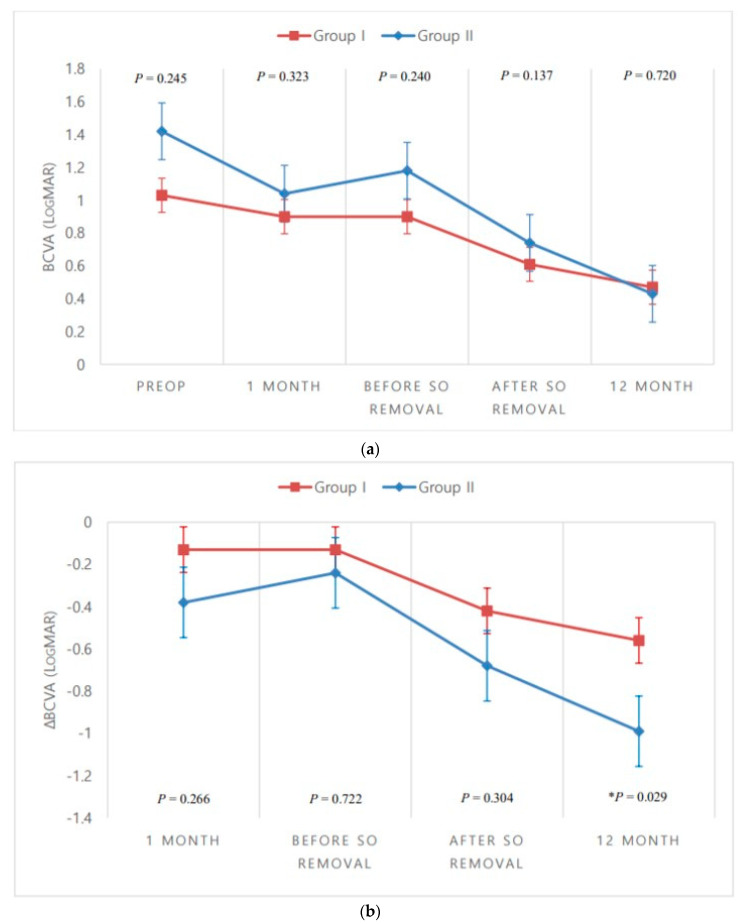
Between-group comparisons over time. (**a**) Best-corrected visual acuity (BCVA); (**b**) Mean change in BCVA from the preoperative (Preop) status. * *p* < 0.05, Mann–Whitney *U*-test. SO, silicone oil.

**Figure 2 jcm-09-03114-f002:**
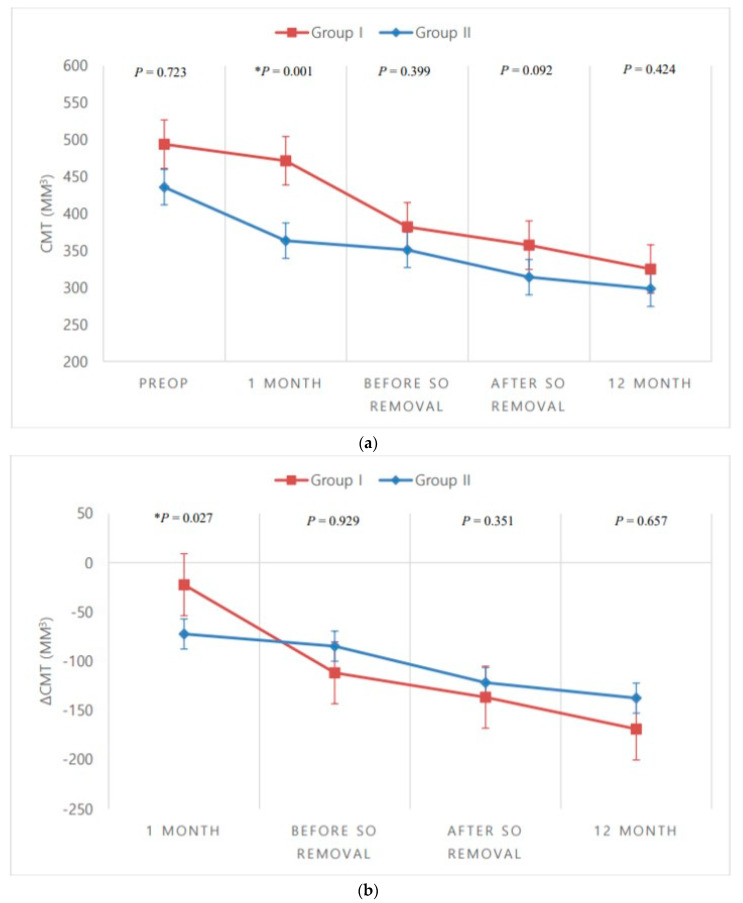
Between-group comparisons over time. (**a**) Mean central macular thickness (CMT); (**b**) Mean change in CMT from the preoperative (Preop) status. * *p* < 0.05, Mann–Whitney *U*-test. SO, silicone oil.

**Table 1 jcm-09-03114-t001:** Patient Demographics and Clinical Characteristics.

	Group I (n = 26)	Group II (n = 20)	*p*-Value
Age (years)	54.77 ± 4.97	54.60 ± 9.19	0.283 ^1^
Sex (n, %)			
Male	14 (53.8%)	12 (60.0%)	0.454 ^2^
Female	12 (46.2%)	8 (40.0%)	
Laterality (n, %)			
Right	6 (23.1%)	4 (20.0%)	0.547 ^2^
Left	20 (76.9%)	16 (80.0%)	
HTN (n, %)	8 (30.8%)	4 (20.0%)	0.316 ^2^
Duration of DM (years)	7.60 ± 6.02	5.80 ± 7.90	0.074 ^1^
Preop hemoglobin A1C (%)	8.17 ± 0.91	9.56 ± 2.61	0.022 ^1^
PRP (n, %)	14 (53.8%)	8 (40.0%)	0.263 ^2^
Preop IOP (mmHg)	15.25 ± 3.34	17.19 ± 6.41	0.739 ^1^
Preop BCVA (logMAR [Snellen])	1.03 ± 0.69 [20/214]	1.42 ± 1.03 [20/526]	0.245 ^1^
Preop CMT (μm)	493.92 ± 219.63	435.80 ± 125.22	0.723 ^1^
Preop IVB injections (n)	1.40 ± 0.82	1.08 ± 0.27	0.172 ^1^
Postop hemoglobin A1C (%)	7.89 ± 1.76	6.94 ± 0.33	0.328 ^1^
Combine cataract surgery (n, %)	16 (61.5%)	16 (80.0%)	0.153 ^2^
SO volume (cc)	4.22 ± 0.31	4.46 ± 0.60	0.368 ^1^
Duration of SO (months)	4.46 ± 2.70	4.00 ± 1.72	0.500 ^1^

Values are presented as the mean ± standard deviation, or a number (%). Group I received an SO tamponade without IVB injection; group II received an SO tamponade with IVB injection; HTN, hypertension; DM, diabetes mellitus; Preop, preoperative; PRP, panretinal photocoagulation; IOP, intraocular pressure; BCVA, best-corrected visual acuity; logMAR, logarithm of the minimum angle of resolution; CMT, central macular thickness; IVB, intravitreal bevacizumab injection; SO, silicone oil; Postop, postoperative. ^1^ Mann–Whitney *U*-test; ^2^ Fisher’s exact test.

**Table 2 jcm-09-03114-t002:** Between-Group Comparisons of Postoperative Clinical Outcomes.

	Group I ^1^ (n = 26)	Group II ^1^ (n = 20)	*p*-Value ^2^
BCVA (logMAR [Snellen])			
Preop	1.03 ± 0.69 [20/214]	1.42 ± 1.03 [20/526]	0.245
POD 1 month	0.90 ± 0.67 [20/159]	1.04 ± 0.61 [20/219]	0.323
Before SO removal	0.90 ± 0.46 [20/159]	1.18 ± 0.66 [20/303]	0.240
1 month after SO removal	0.61 ± 0.46 * [20/81]	0.74 ± 0.46 * [20/110]	0.137
POD 12 month	0.47 ± 0.51 * [20/59]	0.43 ± 0.52 * [20/54]	0.720
ΔBCVA (logMAR)			
POD 1 month	−0.13 ± 0.77	−0.38 ± 0.88	0.266
Before SO removal	−0.13 ± 0.66	−0.24 ± 0.76	0.722
After SO removal	−0.42 ± 0.72	−0.68 ± 0.77	0.304
POD 12 month	−0.56 ± 0.80	−0.99 ± 0.73	0.029
CMT (μm)			
Preop	493.92 ± 219.63	435.80 ± 125.22	0.723
POD 1 month	471.54 ± 120.14	363.40 ± 59.57 *	0.001
Before SO removal	382.08 ± 73.63	351.00 ± 41.17 *	0.399
1 month after SO removal	357.31 ± 84.21 *	314.00 ± 38.53 *	0.092
POD 12 month	325.00 ± 77.71 *	298.30 ± 29.58 *	0.424
ΔCMT (μm)			
POD 1 month	−22.39 ± 203.99	−72.40 ± 139.35	0.027
Before SO removal	−111.85 ± 202.61	−84.80 ± 139.98	0.929
After SO removal	−136.62 ± 243.70	−121.80 ± 135.81	0.351
POD 12 month	−168.92 ± 239.76	−137.50 ± 132.67	0.657

Values are presented as the mean ± standard deviation unless otherwise indicated. Group I received an SO tamponade without intravitreal bevacizumab injection; group II received an SO tamponade with intravitreal bevacizumab injection; BCVA, best-corrected visual acuity; logMAR, logarithm of the minimum angle of resolution; Preop, preoperative; POD, postoperative day; SO, silicone oil; CMT, central macular thickness. ^1^ Wilcoxon signed-rank test. ^2^ Mann–Whitney *U*-test. ***
*p* < 0.05 compared to preoperative measurement.

**Table 3 jcm-09-03114-t003:** Postoperative Complications and Other Outcomes.

	Group I (n = 26)	Group II (n = 20)	*p*-Value
Postop complication	3 (11.5%)	2 (10.0%)	0.627 ^1^
IOP increase	0	2	
Vitreous hemorrhage	1	0	
Fibrovascular membrane proliferation	2	0	
12-month Postop hemoglobin A1c (%)	7.89 ± 1.76	6.94 ± 0.33	0.328 ^2^
IVB injections after SO removal (n, %)	6 (23.1%)	0 (0.0%)	0.025 ^1^

Values are presented as the mean ± standard deviation or a number (%). Group I received an SO tamponade without IVB injection; group II received an SO tamponade with IVB injection; IOP, intraocular pressure; Postop, postoperative; SO, silicone oil; IVB, intravitreal bevacizumab. ^1^ Fisher’s exact test; ^2^ Mann–Whitney U-test.

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
