# Peer review of "Effect of Intrasilicone Bevacizumab Injection in Diabetic Tractional Retinal Detachment Surgery: A Retrospective Case-Control Study"

_jcm, 2020, doi:10.3390/jcm9103114_

Round 1
Reviewer 1 Report
The authors well presented the study. Introduction, Materials, Results and Conclusion paragraph are well written and structured. Images and table are interesting and summarize the results in an endearing way. I have only minor concerns:
- the last phrase of the introduction: "We discovered that intrasilicone bevacizumab injection immediately after vitrectomy rapidly reduced central macular thickness and increased final visual acuity." I think that present immediately the study result is not appropriate.
- "Mean logMAR VA exhibited statistically significant improvement 1 month after SO removal and 12 months after vitrectomy in both groups. Although the mean change of logMAR VA was statistically significantly larger in group II 12 months after surgery, such a between-group comparison is not expected to be of clinical significance due to the variation in cataracts between patients. In the present study, 61.5% of patients in group I and 80.0% of patients in group II underwent cataract removal during the vitrectomy, several patients underwent cataract surgery at the time of SO removal, and we did not exclude patients with preoperative pseudophakia from the study. This variation in cataracts between patients is one of the limitations in our study, as it may have led to selection bias." Maybe the explanation of BCVA changes is a bit simplistic and can be improved
Author Response
- the last phrase of the introduction: "We discovered that intrasilicone bevacizumab injection immediately after vitrectomy rapidly reduced central macular thickness and increased final visual acuity." I think that present immediately the study result is not appropriate.
- As instructed, the last text of the introduction was deleted because it was not in an appropriate position.
- "Mean logMAR VA exhibited statistically significant improvement 1 month after SO removal and 12 months after vitrectomy in both groups. Although the mean change of logMAR VA was statistically significantly larger in group II 12 months after surgery, such a between-group comparison is not expected to be of clinical significance due to the variation in cataracts between patients. In the present study, 61.5% of patients in group I and 80.0% of patients in group II underwent cataract removal during the vitrectomy, several patients underwent cataract surgery at the time of SO removal, and we did not exclude patients with preoperative pseudophakia from the study. This variation in cataracts between patients is one of the limitations in our study, as it may have led to selection bias." Maybe the explanation of BCVA changes is a bit simplistic and can be improved
- Yes. Variables for cataracts have always been considered while writing article, and it is true that variable control is difficult. Further explanation is needed for the results of the study on visual acuity in this paper, and additional explanations are provided after line 226.
As you pointed out, I have revised the paper as much as possible, and thank you very much for reviewing the article.
Reviewer 2 Report
The work of Seung Kook Baek is an interesting one in the field of treatment of Diabetic Tractional Retinal Detachment Surgery, however, the below the concerns needs to be addressed :
Major:
Plagiarism check showed a level of 33%, which is quite high. To be considered for publication, authors need to make a thorough revision to keep the Similarity Index ≤ 20%.
Line 48: Although the purpose is clear, the hypothesis of the study is lacking. The authors need to write a clear hypothesis justifying their purpose.
Lines 54-90: The methods section is not organized and needs to subsectioned such as patient patients recruitment and data collection, surgical technique, statistical analysis,... Authors are encouraged to reformat this part of the methods.
Lines 56-58: These lines should fall under the subsection Ethics Statement.
Line 87: The authors have conducted their study on a limited sample size... Did they try to carry out any sampling/experimental design before conducting the study? Determining the optimal sample size in the statistical analysis study could provide readers about the adequate number of participants needed to detect robust significant results!
Lines 132, 133, 145-149: The authors cite the exact values found in the text (-111.85 ± 202.61, etc) despite being displayed in Table 1. The authors are encouraged to decrease this overlap with the Table and instead summarize these findings. Readers can retrieve this information from the Table...
Line 177 the authors have discussed the role of VEGF in increasing the response to surgical trauma and inflammation in DR patients, however, this is also true in healthy individuals; Gorenjak V et al, Peripheral blood mononuclear cells extracts VEGF protein levels and VEGF mRNA: Associations with inflammatory molecules in a healthy population. PLoS ONE. 2019 Aug 16;14(8). The authors are encouraged to discuss this issue. it will add more insights.
Minor:
The quality of the graphs in figure 1 is poor. The authors are encouraged to rectify this issue…
Author Response
Plagiarism check showed a level of 33%, which is quite high. To be considered for publication, authors need to make a thorough revision to keep the Similarity Index ≤ 20%.
- I modified it as much as possible
Line 48: Although the purpose is clear, the hypothesis of the study is lacking. The authors need to write a clear hypothesis justifying their purpose.
- As you pointed out, content has been added to clarify the purpose of the paper.
Lines 54-90: The methods section is not organized and needs to subsectioned such as patient patients recruitment and data collection, surgical technique, statistical analysis,... Authors are encouraged to reformat this part of the methods.
- I modified it as the reviewer pointed out.
Lines 56-58: These lines should fall under the subsection Ethics Statement.
- I modified it as the reviewer pointed out.
Line 87: The authors have conducted their study on a limited sample size... Did they try to carry out any sampling/experimental design before conducting the study? Determining the optimal sample size in the statistical analysis study could provide readers about the adequate number of participants needed to detect robust significant results!
- It is true that the sample size of this study is small, so there are limitations. In general, we do not see many PDR patients with TRD in our university hospital. In addition, even if TRD surgery is performed, there are cases in which preop OCT cannot be taken because of VH. In some cases, other hospitals have already undergone other surgery, and even if the surgery is performed, it is not easy to recruit because TRD is not involved macula and silicone oil is not injected. I don't think we need to describe sampling or experimental design in our retrospective study, and I think the statistical results of this study's non-parametric statistics were also significant. I am sorry that I did not understand what you pointed out and could not give you the answer you want. Please give me some helpful advice and I will revise it again. Thank you.
Lines 132, 133, 145-149: The authors cite the exact values found in the text (-111.85 ± 202.61, etc) despite being displayed in Table 1. The authors are encouraged to decrease this overlap with the Table and instead summarize these findings. Readers can retrieve this information from the Table...
- I modified it as the reviewer pointed out.
Line 177 the authors have discussed the role of VEGF in increasing the response to surgical trauma and inflammation in DR patients, however, this is also true in healthy individuals; Gorenjak V et al, Peripheral blood mononuclear cells extracts VEGF protein levels and VEGF mRNA: Associations with inflammatory molecules in a healthy population. PLoS ONE. 2019 Aug 16;14(8). The authors are encouraged to discuss this issue. it will add more insights.
- I modified it as the reviewer pointed out.
Minor:
The quality of the graphs in figure 1 is poor. The authors are encouraged to rectify this issue…
- I modified it as the reviewer pointed out.
As you pointed out, I have revised the paper as much as possible, and thank you very much for reviewing the article.
"Please see the attachment"
Best regards,
Seung kook Baek.

Round 2
Reviewer 2 Report
The authors have answered the majority of my comments.
Author Response
Thank you for the paper review, and thank you for making efforts to improve the quality of the paper even though it is difficult due to COVID-19.
King regards
Seung Kook Baek, MD